# Corrosion Behavior of CoCrFeNiTa_x_ Alloys in 1 M Sodium Chloride Aqueous Solution

**DOI:** 10.3390/ma13225157

**Published:** 2020-11-16

**Authors:** Chun-Huei Tsau, Rong-Wei Hsiao, Tien-Yu Chien

**Affiliations:** Institute of Nanomaterials, Chinese Culture University, Huagang Road, 55, Shilin District, Taipei 11114, Taiwan; chiji0932686087@gmail.com (R.-W.H.); school0952067023@gmail.com (T.-Y.C.)

**Keywords:** CoCrFeNiTa_x_ alloys, corrosion, microstructures, hardness, polarization test, electrochemical impedance spectroscopy, critical pitting temperature

## Abstract

This paper investigates the effects of Ta content on the microstructures, hardness and corrosion behavior of as-cast CoCrFeNiTa_x_ alloys. The results indicate that the addition of Ta can change the microstructures of these alloys to dual-phased structures (FCC + HCP), as well as increasing their hardness. This study uses constant galvanostatic/potentiometric methods to measure the polarization curves of CoCrFeNiTa_0.1_, CoCrFeNiTa_0.3_ and CoCrFeNiTa_0.5_ alloys in deaerated 1 M sodium chloride solution at different temperatures. Electrochemical impedance spectroscopy is also used to analyze these alloys in sodium chloride solution. The results indicate that the CoCrFeNiTa_0.5_ alloy has a eutectic structure and the highest hardness. Furthermore, although the CoCrFeNiTa_0.5_ alloy has the best corrosion resistance, the CoCrFeNiTa_0.3_ alloy has the best pitting resistance among these alloys.

## 1. Introduction

The high-entropy alloy (HEA) concept provides a smart method for the design of suitable alloys for diverse applications [1,2,3]. Yeh points out that there are four core features associated with HEAs, namely, high entropy, sluggish diffusion, severe lattice distortion and cocktail effects [4], which impart them with unique properties. The HEA concept is widely used to design alloys for mechanical, chemical and physical applications [5,6,7], and is further extended to include high-entropy ceramics [8,9]. This means that the HEA concept can be widely used to develop new materials for various applications.

Corrosion is always a major problem in the application of metals, and research into how to enhance the corrosion resistance of metals is important. The HEA concept can be used to select appropriate elements to produce new alloys with good corrosion resistance [10,11]. The corrosion resistance of alloys can also be enhanced through surface coating with HEAs [12,13,14,15]. Thermally-sprayed HEAs can effectively improve the corrosion resistance of materials. AlCoCr_x_FeNi and Al_2_CrFeCo_x_CuNiTi alloys were coated on the surface of steel and improved corrosion resistance in NaCl solutions. CoCrFeNi equiatomic alloy has an FCC granular structure and some HCP-structured, Cr-rich precipitates in the matrix [16]. Many CoCrFeNi-based HEAs have been investigated with regard to variations in their microstructures and mechanical and chemical properties in order to evaluate their potential applications. Adding niobium or molybdenum into CoCrFeNi alloys can change their microstructures into dual-phased structures; the corrosion resistance of CoCrFeNi(Nb,Mo) alloys has been found to be reduced after Nb or Mo addition, and the hardness of these alloys increased significantly [17]. CoCrFeMnNi alloy retains an FCC granular structure after adding manganese [18]. This alloy can be enhanced by adding Nb or Ti due to the formation of Laves phases [19]. Experiments on CoCrFeNiHf_x_ alloys showed similar results: the microstructures of CoCrFeNiHf_x_ alloys changed to dual-phased microstructures (FCC and Laves phases) and their hardness increased [20]. The microstructures and mechanical properties of CoCrFeNiTa_x_ alloys have also been reported. Increasing the levels of Ta content in CoCrFeNiTa_x_ alloys resulted in a transformation of the microstructures of these alloys from hypoeutectic to hypereutectic, whilst the CoCrFeNiTa_0.5_ alloy was shown to have a eutectic structure [21]. The present study investigates the effects of Ta content on the microstructures and corrosion resistance of as-cast CoCrFeNiTa_x_ alloys and relates their microstructures to the corrosion behavior of the alloys in 1 M deaerated sodium chloride solution.

## 2. Materials and Methods

Three experimental alloys, CoCrFeNiTa_0.1_, CoCrFeNiTa_0.3_ and CoCrFeNiTa_0.5_ (named Ta_0.1_, Ta_0.3_ and Ta_0.5_), were prepared by arc melting in an argon atmosphere with the purity of the cobalt, chromium, iron, nickel and tantalum being higher than 99.9%. Each alloy had a total weight of about 100 g, and all were remelted five times. The maximum amount of Ta added to each alloy was half the amount of the other elements, because tantalum has a larger atomic weight, i.e., 180.95 g/mol [22]. The nominal compositions of as-cast CoCrFeNiTa_x_ alloys are listed in Table 1. The structures of the alloys were examined by an X-ray diffractometer (XRD, Rigagu ME510-FM2, Rigaku Ltd., Tokyo, Japan) operated at 30 kV and a scanning rate of 0.04 degrees per second. The source of the XRD was a copper target with a wavelength (CuKα) of 1.5406 Å, and the size of the samples was about 1 cm^2^. The microstructures of the CoCrFeNiTa_x_ alloys were observed using a scanning electron microscope (SEM, JEOL-5410, JEOL Ltd., Tokyo, Japan); the compositions of the alloys were examined by an energy dispersive spectrometer (SEM/EDS), operating at 25 kV. The etching solution was aqua regia, a mixed solution of one part HNO_3_ and three parts HCl. A backscattered electron image (BEI) was used to observe the microstructures; a phase with a higher average atomic number was brighter than a phase with a lower average atomic number because of the effect of backscattering electrons. The hardness of the CoCrFeNiTa_x_ alloys was tested with a Vicker’s hardness tester (Matsuzawa MMT-X3B, Matsuzawa Co., Akita, Japan) under a load of 19.6 N. Each alloy was measured five times to average the hardness.

Both the polarization curves and electrochemical impedance spectroscopy (EIS) of the CoCrFeNiTa_x_ alloys were tested using a potentiostat/galvanostat (Autolab PGSTAT302N, Metrohm Autolab B.V., Utrecht, The Netherlands). Specimens for the polarization test were mounted in epoxy resin and the exposed areas were 0.5 cm in diameter (0.1964 cm^2^). A saturated silver chloride electrode (Ag/AgCl, V_SSE_) was selected as the reference electrode; the reduction potential of saturated silver chloride electrodes is 197 mV higher than that of a standard hydrogen electrode (V_SHE_) scale at 25 °C [23]. A smooth Pt sheet was used as the counter electrode. All of the specimens for the polarization test were mechanically wet polished with 1200 SiC grit paper. The 1 M concentration test solution was prepared from reagent-grade sodium chloride and distilled water. The solution was deaerated by bubbling nitrogen gas through it prior to and during the polarization experiments to eliminate the effect of dissolved oxygen. The polarization test started after the specimen, counter electrodes and reference electrodes had been in the bubbling solution for 900 s. The scanning rate of the polarization curves was 1 mV/s and the test temperatures were 30 °C, 40 °C, 50 °C and 60 °C. The critical pitting temperatures (CPTs) of the CoCrFeNiTa_x_ alloys were measured according to the ASTM G150-99 standard [24]. A schematic diagram of the CPT test system is shown in Figure 1. A power supplier (Keithley 2400, Keithley Instruments, Cleveland, OH, USA) provided the current at a fixed potential. A recorder (Agilent 34970A, Keysight Technologies, Santa Rosa, CA, USA) read the values of the current and temperature every 1 s. The increasing rate of the temperature of the solution was 1 °C/min.

## 3. Results

Figure 2 shows the XRD patterns of the as-cast CoCrFeNiTa_x_ alloys. There are two phases in these alloys, namely FCC and Laves phases (HCP structure). The FCC phase was the main phase for the Ta_0.1_ alloy; only a few HCP-structured particles (Laves phase) were in this alloy. Increasing the Ta content resulted in an increase in the Laves phase, which then became the major phase for the Ta_0.5_ alloy. The lattice constants of the FCC and Laves phases in the alloys are listed in Table 2.

Figure 3 displays the SEM BEI microstructures of the as-cast CoCrFeNiTa_x_ alloys. Both the Ta_0.1_ and Ta_0.3_ alloys showed a hypoeutectic structure. In the Ta_0.1_ alloy, the dendrites formed an FCC phase, and the interdendrites almost formed a single Laves phase, as shown in Figure 3a. In the Ta_0.3_ alloy, the dendrites formed an FCC phase and the interdendrites formed a eutectic structure of FCC and Laves phases, as shown in Figure 3b. The Ta_0.5_ alloy displayed a eutectic microstructure, as shown in Figure 3c. However, a small number of primary dendrites were still observed in the Ta_0.5_ alloy, indicating that the Ta_0.5_ alloy had a hypereutectic structure, as shown in Figure 3d. The overall chemical compositions and the compositions of each phase of these alloys analyzed by SEM/EDS are listed in Table 3. The CoCrFeNi alloy has a granular FCC structure [16], and only a small amount of Ta could be solid-soluted in the FCC phase. Therefore, the level of Ta content in the Laves phases in these alloys was higher than that in the FCC phases. The other elements, Co, Cr, Fe and Ni, had higher levels of Ta content in the FCC phase.

Figure 4 plots the relationship between the hardness of as-cast CoCrFeNiTa_x_ alloys and Ta content. The hardness linearly increases with increasing Ta content in CoCrFeNiTa_x_ alloys. The hardness of the Ta_0.1_ alloy was only 167 HV, while that of Ta_0.5_ reached 467 HV. Adding tantalum into a CoCrFeNi alloy can increase its hardness because tantalum has a larger atomic radius, i.e., 1.43 Å, whilst the atomic radiuses of cobalt, chromium, iron and nickel are 1.253 Å, 1.249 Å, 1.241 Å and 1.243 Å, respectively [22]. Therefore, the hardness of the CoCrFeNiTa_x_ alloys increased because of the solid solution strengthening effect. In addition, the hardness of the HCP-structured Laves phases was higher than that of the FCC phases because the HCP phases presented less slip system. Increasing the amount of the HCP phase, and therefore, of the solid solution strengthening effect, resulted in an increase in the hardness of CoCrFeNiTa_x_ alloys after adding more tantalum.

The polarization curves of as-cast CoCrFeNiTa_x_ alloys tested in 1 M deaerated sodium chloride solution at different temperatures are shown in Figure 5a–c. The corrosion current density (*i*_corr_) and corrosion potential (*E*_corr_) of the curves are listed in Table 4. The *i*_corr_ increased with an increase in the test temperature because of the electrochemical reaction. According to the Arrhenius equation, the relationship between corrosion current density and temperature is *i*_corr_ = *A exp*(−*Q/RT*), where *A* is a temperature-independent constant; *Q* is the activation energy; *R* is the gas constant; and *T* is temperature. Therefore, the corrosion activation energies of these alloys were calculated by plotting *ln i*_corr_ in relation to temperature, as shown in Figure 5d. The Ta_0.3_ alloy had the highest activation energy among these alloys. The values of *E*_corr_ for the Ta_0.1_ alloy were around −0.33 V_SHE_ at test temperatures of 30 to 60 °C. The values of *E*_corr_ for the Ta_0.3_ alloy were around −0.28 V_SHE_ at test temperatures below 50 °C, and the *E*_corr_ increased to −0.21 V_SHE_ as the test temperature reached 60 °C. The values of *E*_corr_ for the Ta_0.5_ alloy were around −0.23 V_SHE_ at test temperatures below 50 °C, and the *E*_corr_ decreased to −0.35 V_SHE_ as the test temperature reached 60 °C. Additionally, the passivation regions of the Ta_0.1_ and Ta_0.5_ alloys broke down when the test temperature reached 50 °C, but the passivation regions of the Ta_0.3_ alloy retained their complete shapes at 30 to 60 °C. This indicates that the Ta_0.3_ alloy has a better pitting resistance in 1 M deaerated sodium chloride solution. The passivation regions of the Ta_0.1_ and Ta_0.5_ alloys broke down at higher temperatures because they could not resist the pitting corrosion of Cl^−^ ions.

Figure 6 shows the Nyquist plots, Bode plots and equivalent circuit diagrams of as-cast CoCrFeNiTa_x_ alloys in 1 M deaerated NaCl solution derived from electrochemical impedance spectroscopy measurements. The start points of the Nyquist plots of these alloys are very close, as shown in Figure 6a. The Nyquist plot of the Ta_0.3_ alloy has a clearly rising tail, indicating that the corrosion was a mass (ion) transport-controlled process at a lower frequency [25]. As the radius of the semicircle increases with increasing levels of Ta content, this means that the polarization resistance (*R*_p_) increases as levels of Ta content increase. The Ta_0.5_ alloy thus had the best corrosion resistance in 1 M deaerated NaCl solution, as shown in Figure 6b. The resistances of the CoCrFeNiTa_x_ alloys are listed in Table 5. The solution resistances (*R*_s_) of these alloys are almost the same because of the use of the same solution. The corresponding equivalent circuit diagrams of the CoCrFeNiTa_x_ alloys are shown in Figure 6c.

Figure 7 shows the critical pitting temperature test results of as-cast CoCrFeNiTa_x_ alloys. Figure 7a–c shows the CPT curves of the Ta_0.1_, Ta_0.3_ and Ta_0.5_ alloys, respectively. Figure 7d plots the CPTs in relation to applied potential. The Ta_0.1_ alloy can be seen to have a poor pitting resistance by comparing these curves. The pitting resistance of Ta_0.5_ alloy was better than that of the Ta_0.1_ alloy. The CPTs of the Ta_0.1_ and Ta_0.5_ alloys significantly decreased as the applied potential increased. This also proves that the passivation regions of the Ta_0.1_ and Ta_0.5_ alloys broke down as test temperatures increased. The Ta_0.3_ alloy had the best pitting resistance from among these alloys and the CPT only slightly decreased from 58.3 to 52.5 °C as the applied potential increased from 700 mV_SHE_ to 900 mV_SHE_.

Figure 8 shows the surface morphologies of as-cast CoCrFeNiTa_x_ alloys after polarization tests in 1 M deaerated NaCl solution at 30 °C. The major phases in these alloys were FCC phases and HCP-structured Laves phases. These images indicate that the FCC phases in these alloys were the main corrosion phases, that is, the FCC phases were the anodes and the Laves phases were the cathodes in these alloys tested in NaCl solution. In addition, the FCC/HCP phase boundaries were also easily corroded because they were defects and thus had higher Gibbs free energies. The Ta_0.5_ alloy had the most interphase boundaries because of its eutectic structure. Therefore, the Ta_0.5_ alloy had a pool pitting resistance at higher temperatures. High temperature accelerated the electrochemical reaction, causing Cl^−^ ions to attack the interface, and thus, the passivation films were broken down. However, the Ta_0.5_ alloy had the best corrosion resistance in 1 M NaCl solution at low temperatures.

## 4. Conclusions

This paper examined the microstructures and hardness of as-cast CoCrFeNiTa_x_ alloys. Some important results are listed below:The phases in as-cast CoCrFeNiTa_x_ alloys were FCC phases and HCP-structured Laves phases. Increasing levels of Ta content in these alloys resulted in increases in the amount of Laves phases. Ta_0.1_ and Ta_0.3_ alloys had hypoeutectic microstructures and the Ta_0.5_ alloy had a eutectic microstructure.The hardness of the as-cast CoCrFeNiTa_x_ alloys increased with increasing levels of Ta content. Increasing the levels of Ta content in these alloys had two effects, namely, solid solution strengthening and increasing amounts of HCP-structured Laves phases. Furthermore, the Laves phases in these alloys had higher levels of Ta content than in FCC phases. The Ta_0.5_ alloy had the highest hardness of 467 HV.Among the CoCrFeNiTa_x_ alloys, the Ta_0.5_ alloy had the best corrosion resistance in 1 M deaerated NaCl solution. The Ta_0.5_ alloy had a polarization resistance of 30.7 kΩ, the highest from among these alloys. However, the Ta_0.5_ alloy had the best pitting resistance in 1 M NaCl solution.


## Figures and Tables

**Figure 1 materials-13-05157-f001:**
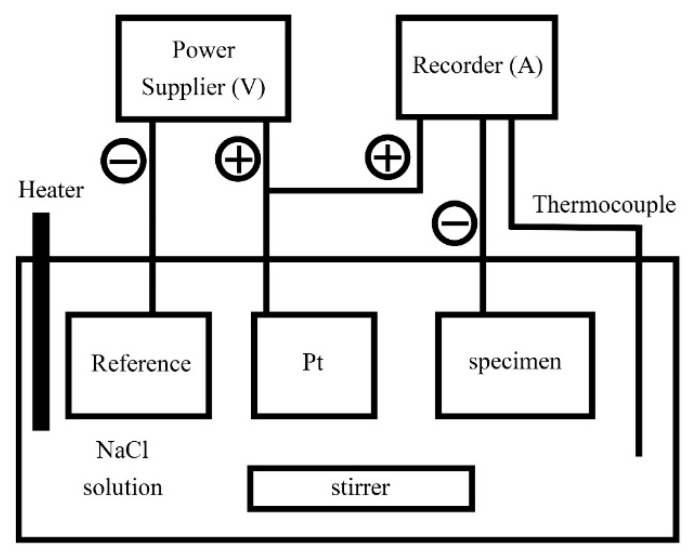
A schematic diagram shows the critical pitting temperature (CPT) test system.

**Figure 2 materials-13-05157-f002:**
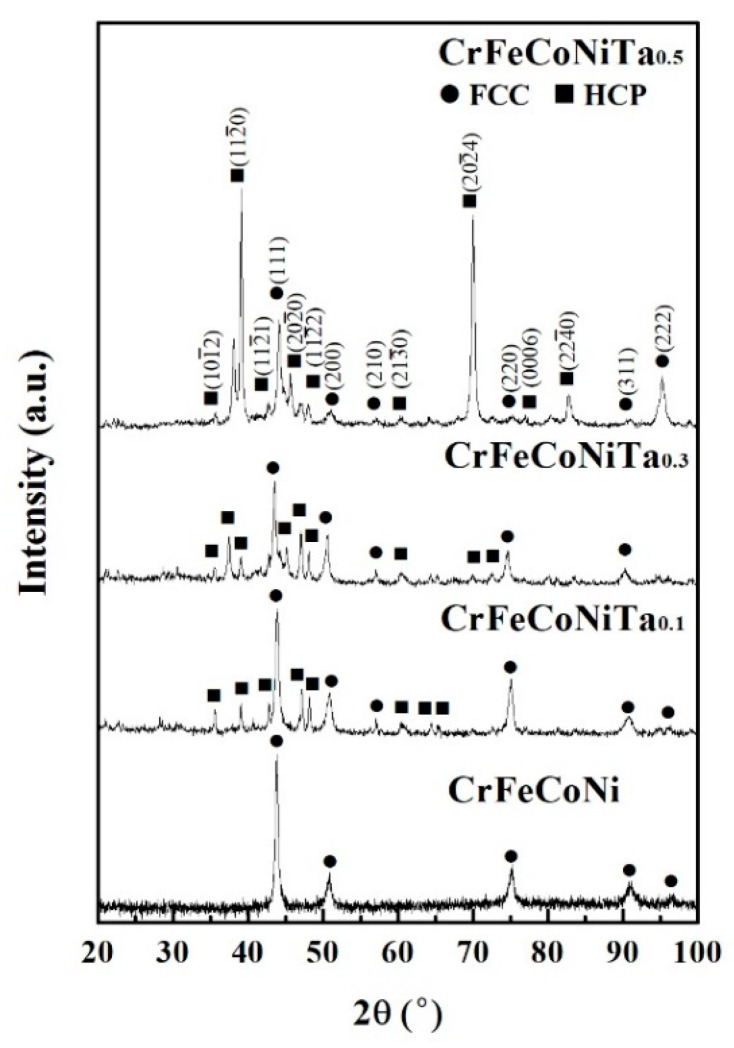
X-ray diffractometer (XRD) patterns of as-cast CoCrFeNiTa_x_ alloys.

**Figure 3 materials-13-05157-f003:**
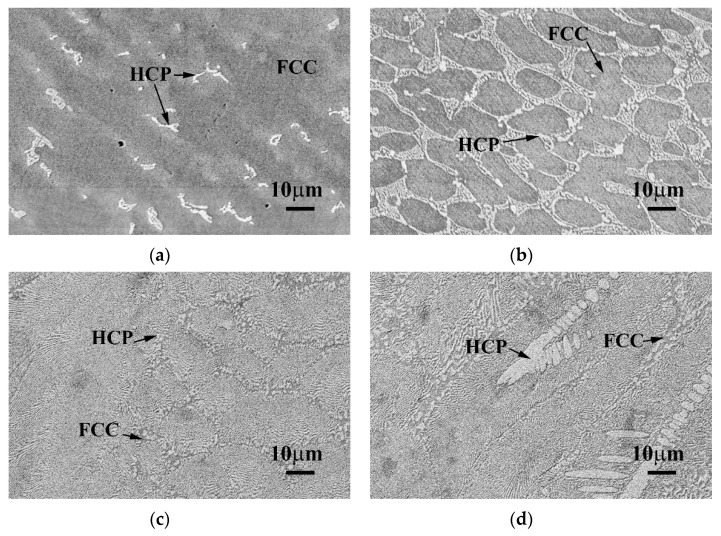
Scanning electron microscope backscattered electron image (SEM BEI) microstructures of the as-cast CoCrFeNiTa_x_ alloys. (**a**) Ta_0.1_ alloy; (**b**) Ta_0.3_ alloy; (**c**) Ta_0.5_ alloy; and (**d**) Ta_0.5_ alloy with some primary HCP dendrites. The brighter phase had a higher average atomic number, i.e., the Ta content in the brighter phase (HCP) was higher than in the darker phase (FCC).

**Figure 4 materials-13-05157-f004:**
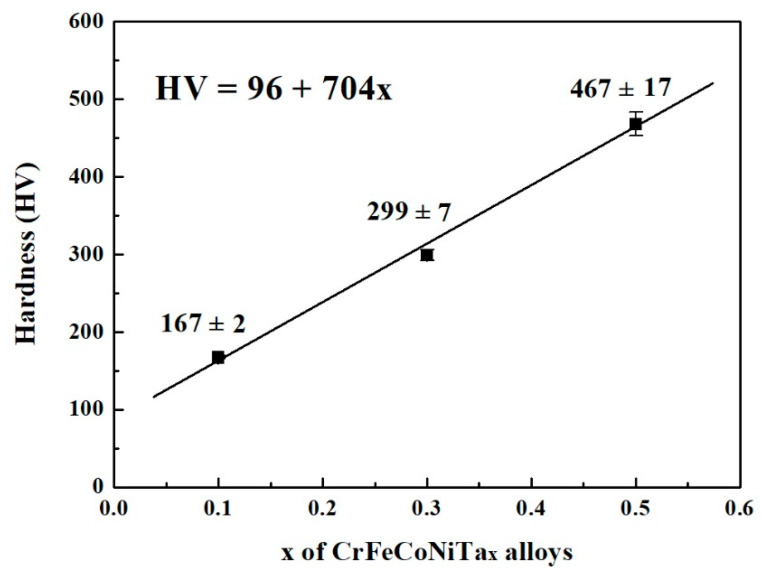
The relationship between the hardness and Ta content of as-cast CoCrFeNiTa_x_ alloys. The load was 19.6 N.

**Figure 5 materials-13-05157-f005:**
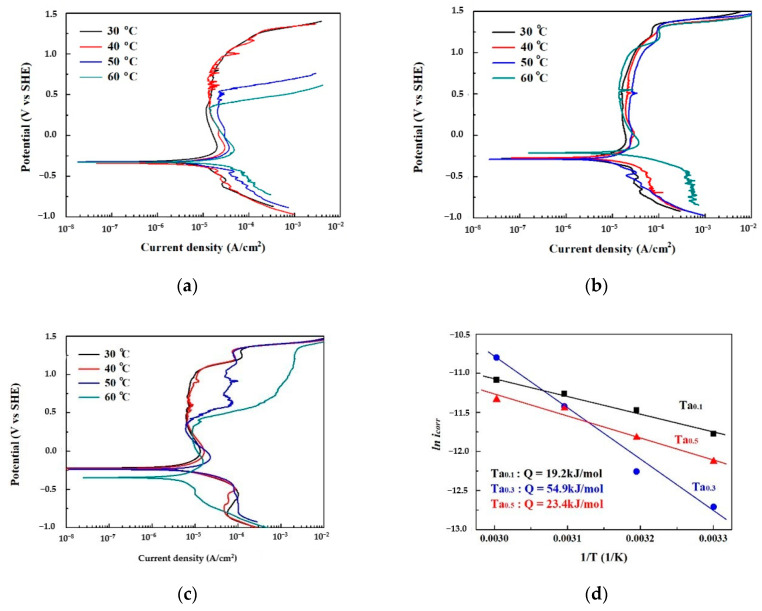
Polarization results of as-cast CoCrFeNiTa_x_ alloys in 1 M aeration NaCl solution at different temperatures. (**a**) Polarization curves of the Ta_0.1_ alloy; (**b**) polarization curves of the Ta_0.3_ alloy; (**c**) polarization curves of the Ta_0.5_ alloy; and (**d**) Arrhenius plot showing the relationship of *i*_corr_ with the temperature of the CoCrFeNiTa_x_ alloys.

**Figure 6 materials-13-05157-f006:**
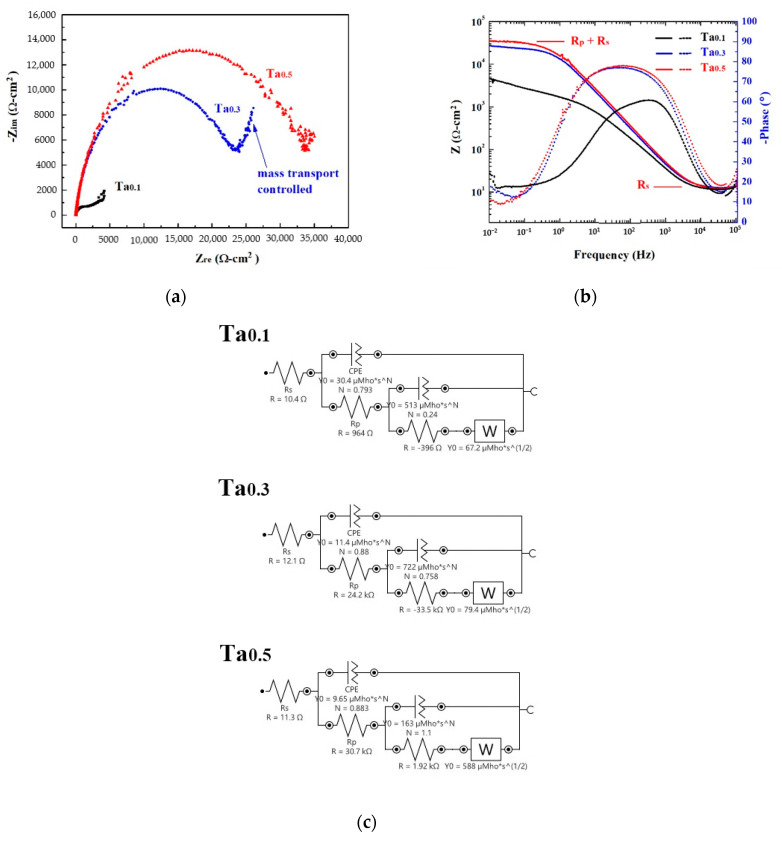
Electrochemical impedance spectroscopy measurements of as-cast CoCrFeNiTa_x_ alloys in 1 M NaCl solution. (**a**) Nyquist plots; (**b**) Bode plots; and (**c**) corresponding equivalent circuit diagrams.

**Figure 7 materials-13-05157-f007:**
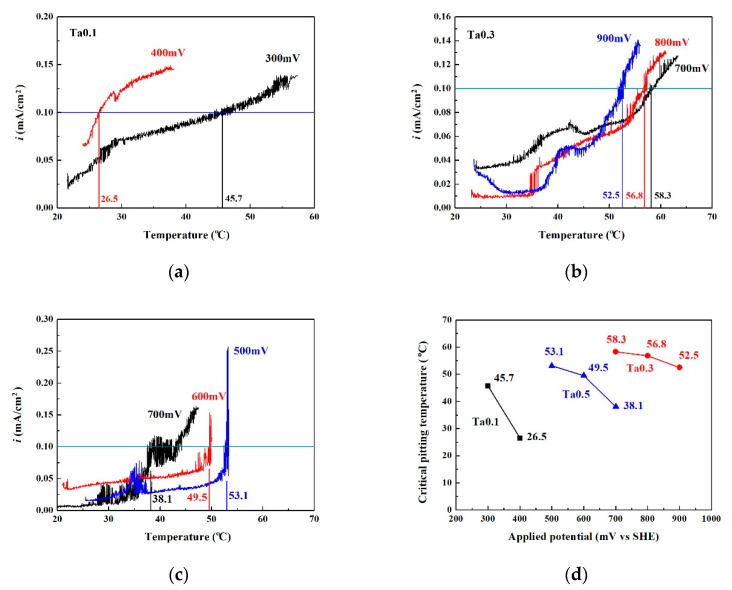
Plots of the CPT test data at different applied potentials (V_SHE_). (**a**) Ta_0.1_ alloy; (**b**) Ta_0.3_ alloy; (**c**) Ta_0.5_ alloy. (**d**) shows the relationship between CPT and applied potential for as-cast CoCrFeNiTa_x_ alloys.

**Figure 8 materials-13-05157-f008:**
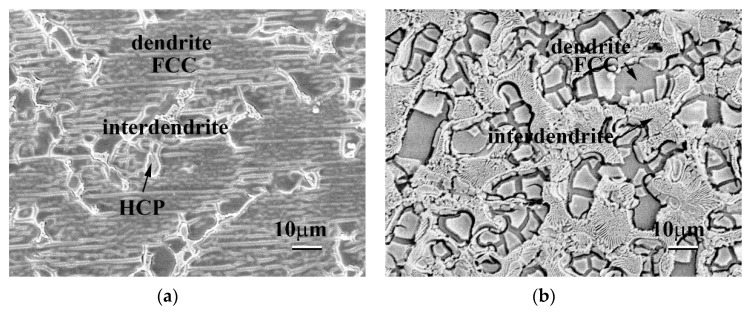
Surface morphologies of as-cast CoCrFeNiTa_x_ alloys after polarization tests in 1 M NaCl solution at 30 °C. (**a**) Ta_0.1_ alloy; (**b**) Ta_0.3_ alloy; and (**c**) Ta_0.5_ alloy.

**Table 1 materials-13-05157-t001:** Nominal compositions of as-cast CoCrFeNiTa_x_ alloys analyzed by scanning electron microscope/energy dispersive spectrometer (SEM/EDS).

Alloys	Compositions (wt.%)				
	Co	Cr	Fe	Ni	Ta
CoCrFeNiTa_0.1_ (Ta_0.1_)	24.2	21.4	22.9	24.1	7.4
CoCrFeNiTa_0.3_ (Ta_0.3_)	21.1	18.6	20.0	21.0	19.3
CoCrFeNiTa_0.5_ (Ta_0.5_)	18.7	16.5	17.7	18.6	28.5

**Table 2 materials-13-05157-t002:** Lattice constants of the phases in the as-cast CoCrFeNiTa_x_ alloys.

Alloys	FCC	HCP	
	*a* (Å)	*a* (Å)	*c* (Å)
Ta_0.1_	3.58	4.61	7.62
Ta_0.3_	3.60	4.61	7.52
Ta_0.5_	3.57	4.61	7.42

**Table 3 materials-13-05157-t003:** Overall chemical compositions and phase compositions (wt.%) of as-cast CoCrFeNiTa_x_ alloys analyzed by SEM/EDS.

Alloys and Phases		Co	Cr	Fe	Ni	Ta
Ta_0.1_	Overall	24.3 ± 0.4	24.3 ± 0.4	24.8 ± 0.6	24.4 ± 0.8	2.3 ± 0.3
	FCC	24.0 ± 0.2	24.8 ± 0.3	25.4 ± 0.5	24.2 ± 0.6	1.8 ± 0.2
	Laves Phase	24.2 ± 0.2	19.0 ± 0.4	17.3 ± 0.6	22.2 ± 0.3	17.4 ± 0.2
Ta_0.3_	Overall	22.8 ± 0.4	25.3 ± 0.4	22.8 ± 0.6	22.8 ± 0.2	6.5 ± 0.3
	FCC	23.4 ± 0.2	26.7 ± 0.6	24.9 ± 0.3	22.1 ± 0.6	3.5 ± 0.4
	Laves Phase	23.3 ± 0.2	18.5 ± 0.4	17.1 ± 0.6	18.1 ± 0.8	23.2 ± 1.1
Ta_0.5_	Overall	23.3 ± 0.6	22.5 ± 0.7	22.8 ± 0.2	21.2 ± 0.5	10.3 ± 0.7
	FCC	22.7 ± 0.6	24.6 ± 0.5	24.6 ± 0.5	23.6 ± 0.2	4.7 ± 0.8
	Laves Phase	23.9 ± 0.4	18.1 ± 0.2	19.2 ± 0.2	17.2 ± 0.2	21.9 ± 0.4

**Table 4 materials-13-05157-t004:** Polarization data of as-cast CoCrFeNiTa_x_ alloys in 1 M deaerated NaCl solution at different temperatures.

Alloys		30 °C	40 °C	50 °C	60 °C
Ta_0.1_	*E*_corr_ (V_SHE_)	−0.32	−0.34	−0.32	−0.33
	*i*_corr_ (µA)	7.69	10.37	12.80	15.34
Ta_0.3_	*E*_corr_ (V_SHE_)	−0.28	−0.27	−0.29	−0.21
	*i*_corr_ (µA)	3.02	4.75	10.92	20.32
Ta_0.5_	*E*_corr_ (V_SHE_)	−0.22	−0.23	−0.24	−0.35
	*i*_corr_ (µA)	5.40	7.32	10.69	11.92

**Table 5 materials-13-05157-t005:** The resistances of as-cast CoCrFeNiTa_x_ alloys in 1 M NaCl solution.

x	Ta_0.1_	Ta_0.3_	Ta_0.5_
R_s_ (Ω)	10.4	12.1	11.3
R_p_ (kΩ)	0.964	24.2	30.7

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
