# Peer review of "Corrosion Behavior of CoCrFeNiTax Alloys in 1 M Sodium Chloride Aqueous Solution"

_materials, 2020, doi:10.3390/ma13225157_

Round 1

Reviewer 1 Report

The paper is presenting mainly the corrosion resistance and structural behaviour of CoCrFeNiTax alloys. The subject presents an increased interest due to the possible applications of such alloys (biomedical, marine, etc.). The experimental methodology seem to be correct and the alloys were studied from different angles, still some points could be improved:

  • The overall aspect of the results presentation seems a little sketchy, as the experimental results and discussions are presented in an expeditive manner. Please expand more or add a separate section for results discussion, including the overall opinion on the different alloy performance comparison.
  • Please add the number of times the alloys were remelted as you used arc melting technique to prepare them.
  • Please add the parameters for X-ray diffraction analyses (angle, source, exposure time) also the database used to determine phases
  • the title of table 1 is not clear. Those are nominal compositions or those are determined by SEM-EDS
  • Please present the name of the software used for corrosion analyses (Tafel built, etc.)
  • The presentation of corrosion test methodology needs to be rearranged as does not follow the steps in order (i.e. "The scanning rate..., after the "The polarisation test started.....). The schematic diagram is important and thank you for shareing.
  • English needs to be revised in some parts of the manuscript (i.e "the laves phase became to a main phase for")
  • In the XRD analyses the phrase "amount of laves phase" is not supported by data offered. In order to approximate the the % of the phase a semi-quantitative evaluation needs to be performed by the RIR method (Reference Intensity Ratio), and results showed in table 2.
  • In table 2 you need to enter the whole alloy name not just Ta0.1, unless you specified somewhere the specimen name.
  • you are already stating in line 83 that the dendrites are FCC. SEM analyses without EBSD determinations is not able to identify phase. One can aproximate the nature of a phase through SEM-EDS but the certainty is given only by TEM analyses. Please refer to possible FCC structure but only after EDS results.
  • Please change the phases with dendritic and interdendritic in table 3 as the type of phases cannot be determined just with EDS. Or you can name the column "suggested phase type" or "possible phase". It can be explained in the text the suggested phase type for the structure indicated in SEM pictures. 
  • EDS results could be discussed a little more in text ( like other elements content in certain phases)
  • SEM pictures are of very low quality and is hard to distinguish the alloy structure. Also you should identify on the pictures the various phases as dendrite, interdendrite, eutectic, etc.
  • The hardness of the materials is usually writen with the type after the number. For ex. 167HV.
  • The hardness increase in the alloy is mainly generated by the increase in Laves phase (which is known to be brittle) and not by the solution strengthening of FCC with Ta. You shout reverse the discussion. Please see the discussion in the referenced article [18].
  • The corrosion resistance investigation of the alloys represents the novelty of the manuscript and in general is well laid out and contain a rich information on the subject. However, the presentation of the results needs to be better organized with clear distinction between the tests types. It would be nice to name the applications that specific tests are critical.
  • The microscopy analyses of the tested samples is insufficient as it shows only the surface of the alloy. In corrosion studies a transversal section analyses is indicated to show phase evolution and concentration at metal/layer interface.
  • SEM pictures again have low resolutions and also need to contain phase labels.
  • The discussion on the corrosion microscopy results thus should be more extensive and specific to each type of tests.

Author Response

The paper is presenting mainly the corrosion resistance and structural behaviour of CoCrFeNiTax alloys. The subject presents an increased interest due to the possible applications of such alloys (biomedical, marine, etc.). The experimental methodology seem to be correct and the alloys were studied from different angles, still some points could be improved:

The overall aspect of the results presentation seems a little sketchy, as the experimental results and discussions are presented in an expeditive manner. Please expand more or add a separate section for results discussion, including the overall opinion on the different alloy performance comparison.

Reply: We modified our manuscript.

Please add the number of times the alloys were remelted as you used arc melting technique to prepare them.

Reply: 5 times, and it was added in the manuscript.

Please add the parameters for X-ray diffraction analyses (angle, source, exposure time) also the database used to determine phases

Reply: The source of XRD is a copper target which wavelength (CuKa) is 1.5406 Å, and scanning rate is 0.04 degree/sec. This is added into the manuscript.

The title of table 1 is not clear. Those are nominal compositions or those are determined by SEM-EDS

Reply: Those are nominal compositions, the actual compositions are listed in Table 3.

Please present the name of the software used for corrosion analyses (Tafel built, etc.)

Reply: The software is built in the machine we used (Autolab PGSTAT302N otentiostat/galvanostat).

The presentation of corrosion test methodology needs to be rearranged as does not follow the steps in order (i.e. "The scanning rate..., after the "The polarisation test started.....). The schematic diagram is important and thank you for shareing.

Reply: It is modified.

English needs to be revised in some parts of the manuscript (i.e "the laves phase became to a main phase for")

Reply: It is modified.

In the XRD analyses the phrase "amount of laves phase" is not supported by data offered. In order to approximate the the % of the phase a semi-quantitative evaluation needs to be performed by the RIR method (Reference Intensity Ratio), and results showed in table 2.

Reply: Because the intensity of HCP phase is higher than that of FCC phase, so this description is correct qualitatively.

In table 2 you need to enter the whole alloy name not just Ta0.1, unless you specified somewhere the specimen name.

Reply: We have specified the specimen name in the section of “Materials and Methods”.

you are already stating in line 83 that the dendrites are FCC. SEM analyses without EBSD determinations is not able to identify phase. One can aproximate the nature of a phase through SEM-EDS but the certainty is given only by TEM analyses. Please refer to possible FCC structure but only after EDS results.

Reply: In the reference “Jiang, H.; Han, K.; Qiao, D.; Lu, Y.; Cao, Z.; Li, T.; Effects of Ta addition on the microstructures and mechanical properties of CoCrFeNi high entropy alloy, Mater. Chem. Phys. 2018, 210, 43-48”, the authors identified the phases already. So we could confirm our data.

Please change the phases with dendritic and interdendritic in table 3 as the type of phases cannot be determined just with EDS. Or you can name the column "suggested phase type" or "possible phase". It can be explained in the text the suggested phase type for the structure indicated in SEM pictures. 

Reply: We could measure the compositions of the phases.

EDS results could be discussed a little more in text (like other elements content in certain phases)

Reply: They were modified.

SEM pictures are of very low quality and is hard to distinguish the alloy structure. Also you should identify on the pictures the various phases as dendrite, interdendrite, eutectic, etc.

Reply: We only have an old conventional SEM. However, the brightness and contrast of the images were adjusted.

The hardness of the materials is usually writen with the type after the number. For ex. 167HV.

Reply: They were modified.

The hardness increase in the alloy is mainly generated by the increase in Laves phase (which is known to be brittle) and not by the solution strengthening of FCC with Ta. You shout reverse the discussion. Please see the discussion in the referenced article [18].

Reply: It is modified.

The corrosion resistance investigation of the alloys represents the novelty of the manuscript and in general is well laid out and contain a rich information on the subject. However, the presentation of the results needs to be better organized with clear distinction between the tests types. It would be nice to name the applications that specific tests are critical.

Reply: We modified our manuscript.

The microscopy analyses of the tested samples is insufficient as it shows only the surface of the alloy. In corrosion studies a transversal section analyses is indicated to show phase evolution and concentration at metal/layer interface.

Reply: It is not included in this manuscript.

SEM pictures again have low resolutions and also need to contain phase labels.

Reply: We only have an old conventional SEM. However, the brightness and contrast of the images were adjusted.

The discussion on the corrosion microscopy results thus should be more extensive and specific to each type of tests.

Reply: We modified our manuscript.

Reviewer 2 Report

The authors have investigated the microstructural, mechanical properties and mainly corrosion properties of Ta containing High entropy alloys and have studied the effect of variation of Ta on these properties. Their study investigates that by increasing the ta content, the microstructure changes to a FCC+HCP dual phase microstructure. The corrosion study indicates that Ta0.5 containing HEA presents better corrosion resistance. I recommend that paper is suitable after minor revision.

The paper needs some more details in the introduction section, specially as authors have talked about the cladding work where HEAs have been used to improve the corrosion resistance like-

Jiang, Y. Q., Li, J., Juan, Y. F., Lu, Z. J., & Jia, W. L. (2019). Evolution in microstructure and corrosion behavior of AlCoCrxFeNi high-entropy alloy coatings fabricated by laser cladding. Journal of Alloys and Compounds775, 1-14.

Argade, G. R., Shukla, S., Liu, K., & Mishra, R. S. (2018). Friction stir lap welding of stainless steel and plain carbon steel to enhance corrosion properties. Journal of Materials Processing Technology259, 259-269.

Qiu, X. W., Wu, M. J., Liu, C. G., Zhang, Y. P., & Huang, C. X. (2017). Corrosion performance of Al2CrFeCoxCuNiTi high-entropy alloy coatings in acid liquids. Journal of Alloys and Compounds708, 353-357.

Please indicate in figure 3 as what they are calling Lave phases, HCP and FCC phase. Figure 3 quality needs to be improved as micron bars are not visible.

How many hardness points per alloy were taken during hardness measurement?

For line 146, what does authors mean by saying that raising tail indicate that corrosion was mass transported. Can they provide any reference for their statement?

It would be good if authors can rewrite the conclusion with highlighting points.  

Author Response

The authors have investigated the microstructural, mechanical properties and mainly corrosion properties of Ta containing High entropy alloys and have studied the effect of variation of Ta on these properties. Their study investigates that by increasing the ta content, the microstructure changes to a FCC+HCP dual phase microstructure. The corrosion study indicates that Ta0.5 containing HEA presents better corrosion resistance. I recommend that paper is suitable after minor revision.

The paper needs some more details in the introduction section, specially as authors have talked about the cladding work where HEAs have been used to improve the corrosion resistance like-

Jiang, Y. Q., Li, J., Juan, Y. F., Lu, Z. J., & Jia, W. L. (2019). Evolution in microstructure and corrosion behavior of AlCoCrxFeNi high-entropy alloy coatings fabricated by laser cladding. Journal of Alloys and Compounds775, 1-14.

Argade, G. R., Shukla, S., Liu, K., & Mishra, R. S. (2018). Friction stir lap welding of stainless steel and plain carbon steel to enhance corrosion properties. Journal of Materials Processing Technology259, 259-269.

Qiu, X. W., Wu, M. J., Liu, C. G., Zhang, Y. P., & Huang, C. X. (2017). Corrosion performance of Al2CrFeCoxCuNiTi high-entropy alloy coatings in acid liquids. Journal of Alloys and Compounds708, 353-357.

Reply: We added some references in our manuscript

Please indicate in figure 3 as what they are calling Lave phases, HCP and FCC phase.

Reply: Figure 3 is modified.

Figure 3 quality needs to be improved as micron bars are not visible.

Reply: We only have an old conventional SEM. However, the brightness and contrast of the images were adjusted.

How many hardness points per alloy were taken during hardness measurement?

Reply: 5 times. It is added in the manuscript.

For line 146, what does authors mean by saying that raising tail indicate that corrosion was mass transported. Can they provide any reference for their statement?

Reply: We add the reference in our manuscript, “Mei, B.A.; Lau, J.; Lin, T.; Tolbert, S.H.; Dunn, B.S.; Pilon, L.; Physical interpretations of electrochemical impedance spectroscopy of redox active electrodes for electrical energy storage, J. Phys. Chem. C 2018, 122, 24499-24511”.

It would be good if authors can rewrite the conclusion with highlighting points.

Reply: We have modified this manuscript.

Reviewer 3 Report

Many thanks for the interesting manuscript.

The introductory part is very short. Individual aspects are only briefly named and not placed in a general context. Especially thermal sprayed coatings referred to in line 32 are only limitedly suitable for corrosion protection applications. The selection appears to be random and generalised. A linkage of the different aspects mentioned is often missing. After the TS HEA coatings, the microstructure of an Cantor-related alloy is directly considered. Additionally, with source 14, this is a self-citation. Therefore a classification in the overall context is mandatory. Then the influence of different alloying elements on corrosion resistance is considered in more detail. However, this has more the character of a listing or designation. A linking discussion and evaluation is missing. The origin of the research object is not deduced from this. The question remains open why the corrosion behaviour for different Ta contents should be considered.

In particular the topic of corrosion properties of Cantor related alloys must be considered in more detail. The motivation for alloy adaptation by refractory metals should be explained and based on this, different development routes should be pointed out, as is already largely the case.

Additional publications have to be included for this purpose. Corresponding literature can be found among others in the following Special Issues

https://www.mdpi.com/journal/entropy/special_issues/High-Entropy_Materials

https://www.mdpi.com/journal/crystals/special_issues/high_entropy_alloy

https://www.mdpi.com/journal/entropy/special_issues/High-Entropy_Alloys

https://www.mdpi.com/journal/coatings/special_issues/HEA-coatings

The experimental part often contains incomplete information on the individual work steps. For example, the exact measurement conditions for phase determination with XRD are missing. In addition, information on the manufacturing conditions is missing. Which sample size was produced? Was a subsequent heat treatment carried out? As the cooling conditions within the castings differ, information on the specimen-taking location is necessary.

In the results section, information must be reconsidered.  The phase assignment in the diffratogram Figure 2 seems doubtful. Here information about the measuring conditions would help. Specify the anode of the X-ray tube (or specify used radiation) Specify measurement parameters: Angle step size, time per step, database used for phase assignment. How were the lattice parameters determined? In some cases very low reflection intensities are assigned. The three diffratograms differ clearly, but the same phases are assigned. Especially the Ta0.5 variant differs significantly from the other two. For example, the reflex at approx. 57.2° is not assigned in the Ta0.5 alloy. Instead, the reflex at approx. 56.1° is assigned to (210). Corresponding corrections have to be made by using the pdf reference. Consequently, the information in Table 2 should also be questioned.

Details of the detector used are to be completed in Fig. 3. In addition, it must be stated whether the images are of etched samples. All in all, the quality of the images is to be rated as rather poor. Fundamental deviations can be shown, but the images are noisy. That the variant Ta0.5 is a eutectic microstructure is disproved by the proportion of larger phase differences. Details of EDX, e.g. voltage used, must be given.

Data on the impedance test is missing (e.g. frequencies, number of measurement repetitions per frequency). Number of measurement repetitions for corrosion test must be indicated.

Additional investigations by means of DSC should be carried out to investigate the exact solidification processes. On the basis of the available database, the conclusions are not justified. 

It is unusual to establish a linear relationship from 3 measuring points. Especially the representation in Figure 4 is critical. An extrapolation of the correlation is to be evaluated even more critically. At least 1 intermediate measure has to be examined in each case to justify the presented correlation.

In the corrosion part a few details have to be added. The fits in the Nyquist plot (or in the Bode plot) should also be shown (i.e. the spectra calculated from the equivalent circuit diagram) Figure 6 c Why was the equivalent circuit diagram taken? References/Explanation?

The values of the resistances etc. are better given in table form - here it is not clear.

In the direct comparison of Figure 3 and 8, it is noticeable that the ratio of the two microstructural components for Ta0.3 has shifted. Thus, significantly larger proportions of the interdendritic region are present in the corrosion sample. This fact must be explained.

Insufficient conclusion, which is only a summary.  On the basis of the investigations carried out, some statements are not proven beyond doubt. Either the statements are to be relativised, additional investigations are to be performed or corresponding literature references are to be included. In the present form a publication is only conceivable with major changes. 

Additional remarks:

Line 55 please write the word hardness in small letters

Line 64 Please check the last sentence

Line 70 insert space before second

Line 72 Please rephrase ...a recorder recorded...

Line 74 Insert space before °C better use K - Generally K should be used instead of °C.

Line 83 Figure… x-axis use 2Theta (°)

Line 84 Figure 2 write thick and insert dot

Author Response

The introductory part is very short. Individual aspects are only briefly named and not placed in a general context. Especially thermal sprayed coatings referred to in line 32 are only limitedly suitable for corrosion protection applications. The selection appears to be random and generalised. A linkage of the different aspects mentioned is often missing. After the TS HEA coatings, the microstructure of an Cantor-related alloy is directly considered. Additionally, with source 14, this is a self-citation. Therefore a classification in the overall context is mandatory. Then the influence of different alloying elements on corrosion resistance is considered in more detail. However, this has more the character of a listing or designation. A linking discussion and evaluation is missing. The origin of the research object is not deduced from this. The question remains open why the corrosion behaviour for different Ta contents should be considered.

Reply: We modified our manuscript, and this manuscript only deals with the corrosion behaviors of high-entropy alloys.

In particular the topic of corrosion properties of Cantor related alloys must be considered in more detail. The motivation for alloy adaptation by refractory metals should be explained and based on this, different development routes should be pointed out, as is already largely the case.

Reply: It is not included in our manuscript.

Additional publications have to be included for this purpose. Corresponding literature can be found among others in the following Special Issues

https://www.mdpi.com/journal/entropy/special_issues/High-Entropy_Materials

https://www.mdpi.com/journal/crystals/special_issues/high_entropy_alloy

https://www.mdpi.com/journal/entropy/special_issues/High-Entropy_Alloys

https://www.mdpi.com/journal/coatings/special_issues/HEA-coatings

Reply: We added some references in this manuscript.

The experimental part often contains incomplete information on the individual work steps. For example, the exact measurement conditions for phase determination with XRD are missing. In addition, information on the manufacturing conditions is missing. Which sample size was produced? Was a subsequent heat treatment carried out? As the cooling conditions within the castings differ, information on the specimen-taking location is necessary.

Reply: The manuscript is modified, we used the as-cast alloys.

In the results section, information must be reconsidered.  The phase assignment in the diffratogram Figure 2 seems doubtful. Here information about the measuring conditions would help. Specify the anode of the X-ray tube (or specify used radiation) Specify measurement parameters: Angle step size, time per step, database used for phase assignment. How were the lattice parameters determined? In some cases very low reflection intensities are assigned. The three diffratograms differ clearly, but the same phases are assigned. Especially the Ta0.5 variant differs significantly from the other two. For example, the reflex at approx. 57.2° is not assigned in the Ta0.5 alloy. Instead, the reflex at approx. 56.1° is assigned to (210). Corresponding corrections have to be made by using the pdf reference. Consequently, the information in Table 2 should also be questioned.

Reply: We remelted this alloy, and tested this alloy again, the strange peak was vanished, only a small peak of FCC (210) was observed. We replotted this figure. Thank you.

Details of the detector used are to be completed in Fig. 3. In addition, it must be stated whether the images are of etched samples. All in all, the quality of the images is to be rated as rather poor. Fundamental deviations can be shown, but the images are noisy. That the variant Ta0.5 is a eutectic microstructure is disproved by the proportion of larger phase differences. Details of EDX, e.g. voltage used, must be given.

Data on the impedance test is missing (e.g. frequencies, number of measurement repetitions per frequency). Number of measurement repetitions for corrosion test must be indicated.

Reply: We only have an old conventional SEM (JEOL-5410), and it was operated at 25 KV.

 Additional investigations by means of DSC should be carried out to investigate the exact solidification processes. On the basis of the available database, the conclusions are not justified. 

Reply: DSC test is not included in this manuscript.

It is unusual to establish a linear relationship from 3 measuring points. Especially the representation in Figure 4 is critical. An extrapolation of the correlation is to be evaluated even more critically. At least 1 intermediate measure has to be examined in each case to justify the presented correlation.

Reply: Only three alloys were in this manuscript, and the data seems to be collinear.

In the corrosion part a few details have to be added. The fits in the Nyquist plot (or in the Bode plot) should also be shown (i.e. the spectra calculated from the equivalent circuit diagram) Figure 6 c Why was the equivalent circuit diagram taken? References/Explanation?

Reply: The figures would become too complex if the fits were added. So we did not put the fitting curves into the figures. The equivalent circuit diagrams were taken from the software of Autolab PGSTAT302N otentiostat/galvanostat (the machine we used).

The values of the resistances etc. are better given in table form - here it is not clear.

Reply: The resistances of these alloys are listed in Table 5.

In the direct comparison of Figure 3 and 8, it is noticeable that the ratio of the two microstructural components for Ta0.3 has shifted. Thus, significantly larger proportions of the interdendritic region are present in the corrosion sample. This fact must be explained.

Reply: All of the images were taken in a SEM, and the magnification was 1000X. Fig.3 was only slight etched, and Fig.8 was heavy etched because of polarization test. We also guest that this difference was caused by sampling, the samples in the Fig.3 and Fig.8 were cut from the different parts of same alloys.

Insufficient conclusion, which is only a summary. On the basis of the investigations carried out, some statements are not proven beyond doubt. Either the statements are to be relativised, additional investigations are to be performed or corresponding literature references are to be included. In the present form a publication is only conceivable with major changes. 

Reply: We modified our manuscript.

Additional remarks:

Line 55 please write the word hardness in small letters

Line 64 Please check the last sentence

Line 70 insert space before second

Line 72 Please rephrase ...a recorder recorded...

Line 74 Insert space before °C better use K - Generally K should be used instead of °C.

Line 83 Figure… x-axis use 2Theta (°)

Line 84 Figure 2 write thick and insert dot

Reply: Some of above have been modified. In our opinion, “Degree (2q)” and “2Theta (°)” are the same, and “°C” is easy to know.

Reviewer 4 Report

Tsau et al. study different aspects of CoCrFeNiTax alloys corrosion behaviour. In particular impact of Ta addition is investigated. This is a very nice contribution which should be published upon minor revision.

(1) What wet chemical methods were used to investigate the alloys' compositions and if not - what is the confidence of chemical composition in bulk?

(2) More explanation on the nature of laves phase would be appreciated, especially targeting a non-expert reader.

(3) Table 2: lattice constants should be quoted with esds.

(4) Table 3: please explain how measurements errors were estimated.

(5) There are still language / editorial mistakes which need correction.

Author Response

Tsau et al. study different aspects of CoCrFeNiTax alloys corrosion behaviour. In particular impact of Ta addition is investigated. This is a very nice contribution which should be published upon minor revision.

(1) What wet chemical methods were used to investigate the alloys' compositions and if not - what is the confidence of chemical composition in bulk?

Reply: We used SEM/EDS to analyze the compositions of the alloys, listed in Table 3.

(2) More explanation on the nature of laves phase would be appreciated, especially targeting a non-expert reader.

Reply: Laves phase is only an alternative name of HCP phase used in High-Entropy Alloys.

(3) Table 2: lattice constants should be quoted with esds.

Reply: Each alloy was examined by XRD twice, and the results were very similar. So the esds are not listed in the table. But the significantly digit of each lattice constant was reduced.

(4) Table 3: please explain how measurements errors were estimated.

Reply: Each value was measured three times to calculate the average.

(5) There are still language / editorial mistakes which need correction.

Reply: We have modified.

Round 2

Reviewer 1 Report

The revised version of the manuscript contains several improvements and I welcome authors reply, but still I believe that the improvements are minor regarding the manuscript as a whole.

To demonstrate this I will compare the issues that were resolved and those that did not receive the necessary attention:

The authors modified the following: number of times the alloy was re-melted (5 times), the specifications for the X-ray analyses ( target type and scanning rate), type of composition on table 1 (nominal), the corrosion software (came with the analyser), corrosion method presentation improved, English revised, the specimens name but not how I expected (should be a separate column in the tabel with specimen name and not like now), the EDS analyses but again not like expected, the images were improved, the hardness convention.  

Unfortunately, there are some points were the authors did not make the suggested revisions. I will list the initial suggestions and the replies bellow:

1. suggestion - The overall aspect of the results presentation seems a little sketchy, as the experimental results and discussions are presented in an expeditive manner.

reply -  we modified the manuscript

The manuscript was not modified in such a way to upgrade from expeditive to detailed . The general aspect remained very close to what was before.

2. In the XRD analyses the phrase "amount of laves phase" is not supported by data 

reply - Because the intensity of HCP phase is higher than that of FCC phase

This answer is not satisfying. The peak height does not give automatically the proportion of the phase. I offered you the name of the method to determine the amount (RIR). Either use this method to determine phase amount in a qualitative matter or not say anything. We should avoid personal approximations. Only the deterministic approach is accepted.

3. you are already stating in line 83 that the dendrites are FCC without EBSD determinations.

reply - In the reference [21], the authors identified the phases already.

When we do an investigation of any type, but especially microscopy, we are not supposed to take the results made by other authors as our results. You are presenting your own results so you should state this accordingly. Again, there is no evidence in your determinations that the dendrites are FCC. The worse part is that you presented the supposition right at the beginning of the SEM results discussion.  It is not absolutely necessary to present the dendrites as FCC and interdendrites as HCP. You can present these statements preferably after the EDS linked to the results of other authors such as [21], but very careful. For example: The EDS results for the dendrite phase show a possible  FCC  structure, also found by other authors [21]. So you can use this information in the hardness and corrosion analyses.

4. Please change the phases with dendritic and interdendritic in table 3 as the type of phases cannot be determined just with EDS. 

reply - We could measure the compositions of the phases.

The EDS analyses gives qualitative information on phase composition and cannot reveal the type of phase. It is not profesional to indicate this in the EDS results. But, in order to advance to the corrosion analyses discussion you can assume that you have FCC and HCP type phases by including a phrase in the text as I showed before.

5. you should identify on the pictures the various phases as dendrite, interdendrite, eutectic, etc.

reply - the labels were added 

They were added indeed and thank you for this but with wrong labels. As i said before you have no way to know that there is a phase FCC and HCP. Please change with DR and ID. Please read more papers and see if SEM images have labeled the type of phases. You can use E for eutectic as this can be observed over MO or SEM.

6. The hardness increase in the alloy is mainly generated by the increase in Laves phase (which is known to be brittle) and not by the solution strengthening of FCC with Ta.

reply - it is modified

Unfortunatelly, it was not modified. I cite: "Adding tantalum into CoCrFeNi alloy can increase the hardness because tantalum has a larger atomic radius 128 of 1.43 Å, and the atomic radiuses of cobalt, chromium, iron and nickel are 1.253, 1.249, 1.241 and 1.243 Å, respectively [22]. Therefore, the hardness of the CoCrFeNiTax alloys increased because of solid solution strengthening effect. In addition, the hardness of HCP-structured laves phase was higher than that of FCC phase because HCP phase had less slip system."

The statement should be like this : "The hardness of the CoCrFeNiTax alloys increased mainly due to the significant presence of the  HCP phase, which is known to be a brittle phase." And than : " In addition to this, the presence of Ta in the FCC phase could lead to solution strengthening, due to it's larger atomic radius." The reader should know the atomic radius of the elements.

7. the presentation of the corrosion results needs to be better organized with clear distinction between the tests types. 

reply -We modified our manuscript.

To me honest I did not see modifications in terms of presentation of corrosion results. I referred to a separation of the results with a subtitle, as they identify separate corrosion problems. However, it is not absolutely necessary if the authors do not consider it important.

8.  In corrosion studies a transversal section analyses is indicated to show phase evolution and concentration at metal/layer interface.

reply - It is not included in this manuscript.

Obviously, it is not included in the manuscript, but why? It should be the corrosion analyses of the alloy as the title says. Is there another manuscript with the same alloy for corrosion analyses? It should raise the paper value considerably. However, it is not absolutely required if authors do not have the investigation.

 In addition to this unanswered issues, I found some uncommon wording: "significantly eutectic structure" , "almost eutectic microstructure" and "almost linearly increased". These word combinations are not ideal and are not found in papers from respected journals. I suggest different wording : "  eutectic-like structure". You have no way to know that there is a linear increase in hardness and is not necessary, as does not present a fitting equation to model the variation. Please extract the word linear from text and the line from graph. Just an increase in hardness is sufficient. 

Author Response

The revised version of the manuscript contains several improvements and I welcome authors reply, but still I believe that the improvements are minor regarding the manuscript as a whole.

To demonstrate this I will compare the issues that were resolved and those that did not receive the necessary attention:

The authors modified the following: number of times the alloy was re-melted (5 times), the specifications for the X-ray analyses ( target type and scanning rate), type of composition on table 1 (nominal), the corrosion software (came with the analyser), corrosion method presentation improved, English revised, the specimens name but not how I expected (should be a separate column in the tabel with specimen name and not like now), the EDS analyses but again not like expected, the images were improved, the hardness convention.  

Unfortunately, there are some points were the authors did not make the suggested revisions. I will list the initial suggestions and the replies bellow:

  1. suggestion - The overall aspect of the results presentation seems a little sketchy, as the experimental results and discussions are presented in an expeditive manner.

reply -  we modified the manuscript

The manuscript was not modified in such a way to upgrade from expeditive to detailed . The general aspect remained very close to what was before.

*Reply: This question is very distinct. Please point out the deficiency exactly. In our opinion, the descriptions of this manuscript is suitable for the readers in this field.

  1. In the XRD analyses the phrase "amount of laves phase" is not supported by data 

reply - Because the intensity of HCP phase is higher than that of FCC phase

This answer is not satisfying. The peak height does not give automatically the proportion of the phase. I offered you the name of the method to determine the amount (RIR). Either use this method to determine phase amount in a qualitative matter or not say anything. We should avoid personal approximations. Only the deterministic approach is accepted.

*Reply: We modified the sentences, Lines 95-96.

  1. you are already stating in line 83 that the dendrites are FCC without EBSD determinations.

reply - In the reference [21], the authors identified the phases already.

When we do an investigation of any type, but especially microscopy, we are not supposed to take the results made by other authors as our results. You are presenting your own results so you should state this accordingly. Again, there is no evidence in your determinations that the dendrites are FCC. The worse part is that you presented the supposition right at the beginning of the SEM results discussion.  It is not absolutely necessary to present the dendrites as FCC and interdendrites as HCP. You can present these statements preferably after the EDS linked to the results of other authors such as [21], but very careful. For example: The EDS results for the dendrite phase show a possible  FCC  structure, also found by other authors [21]. So you can use this information in the hardness and corrosion analyses.

*Reply: CoCrFeNi alloy had an FCC granular structure which was identified in our previously study. After added small amount of Ta into CoCrFeNi alloy, CoCrFeNiTa0.1 alloy had two phases which were FCC and HCP phases. The intensity of the peaks of HCP was very low by comparing with that of FCC phase, this indicated that the HCP phase was less than the FCC phase in Ta0.1 alloy. The SEM BEI image of Ta0.1 alloy also displayed a few brighter particles in the matrix, which were the new phase. Therefore, we could “actually” point out the phases existing the matrix by comparing with the XRD patterns and SEM BEI images, not “possible”. This is the basic logic and method of materials analysis for a materials researcher. In the reference [21], the authors studied the CoCrFeNiTax alloys and described that there were only two phases (FCC and HCP) existing in these alloys, this reference also confirms our results.

Why can’t we use the reference to confirm our data? Or what we need the references for?

  1. Please change the phases with dendritic and interdendritic in table 3 as the type of phases cannot be determined just with EDS. 

reply - We could measure the compositions of the phases.

The EDS analyses gives qualitative information on phase composition and cannot reveal the type of phase. It is not profesional to indicate this in the EDS results. But, in order to advance to the corrosion analyses discussion you can assume that you have FCC and HCP type phases by including a phrase in the text as I showed before.

*Reply: As described above, we can make sure to the phases in these alloys. We also studied the CoCrFeNiNb and CoCrFeNiNb0.5Mo0.5 alloys, and they had similar results.

  1. you should identify on the pictures the various phases as dendrite, interdendrite, eutectic, etc.

reply - the labels were added 

They were added indeed and thank you for this but with wrong labels. As i said before you have no way to know that there is a phase FCC and HCP. Please change with DR and ID. Please read more papers and see if SEM images have labeled the type of phases. You can use E for eutectic as this can be observed over MO or SEM.

*Reply: We can point out the phases existing in these alloys by materials analysis as described above.

  1. The hardness increase in the alloy is mainly generated by the increase in Laves phase (which is known to be brittle) and not by the solution strengthening of FCC with Ta.

reply - it is modified

Unfortunatelly, it was not modified. I cite: "Adding tantalum into CoCrFeNi alloy can increase the hardness because tantalum has a larger atomic radius 128 of 1.43 Å, and the atomic radiuses of cobalt, chromium, iron and nickel are 1.253, 1.249, 1.241 and 1.243 Å, respectively [22]. Therefore, the hardness of the CoCrFeNiTax alloys increased because of solid solution strengthening effect. In addition, the hardness of HCP-structured laves phase was higher than that of FCC phase because HCP phase had less slip system."

The statement should be like this : "The hardness of the CoCrFeNiTax alloys increased mainly due to the significant presence of the  HCP phase, which is known to be a brittle phase." And than : " In addition to this, the presence of Ta in the FCC phase could lead to solution strengthening, due to it's larger atomic radius." The reader should know the atomic radius of the elements.

*Reply: We do not know what your problem is. The similar statements are in Line 125-134, also describes the radius of Ta and the other elements. The reason of hardness increasing have been also emphasized in the second conclusion. We have done it already.

  1. the presentation of the corrosion results needs to be better organized with clear distinction between the tests types. 

reply -We modified our manuscript.

To me honest I did not see modifications in terms of presentation of corrosion results. I referred to a separation of the results with a subtitle, as they identify separate corrosion problems. However, it is not absolutely necessary if the authors do not consider it important.

*Reply: We also do not know what your problem is. The readers whose investigation is in this field should know the descriptions in our manuscript.

  1. In corrosion studies a transversal section analyses is indicated to show phase evolution and concentration at metal/layer interface.

reply - It is not included in this manuscript.

Obviously, it is not included in the manuscript, but why? It should be the corrosion analyses of the alloy as the title says. Is there another manuscript with the same alloy for corrosion analyses? It should raise the paper value considerably. However, it is not absolutely required if authors do not have the investigation.

*Reply: It is easily to understand where the corroded areas are form the SEM image. And the depth of corrosion will change with the parameters of polarization, such as time, temperature and applied potential. We just want to know which phase is anode, and this does not need the transversal section image.

 In addition to this unanswered issues, I found some uncommon wording: "significantly eutectic structure" , "almost eutectic microstructure" and "almost linearly increased". These word combinations are not ideal and are not found in papers from respected journals. I suggest different wording : "  eutectic-like structure". You have no way to know that there is a linear increase in hardness and is not necessary, as does not present a fitting equation to model the variation. Please extract the word linear from text and the line from graph. Just an increase in hardness is sufficient. 

*Reply: We delete the words of “significantly: and “almost”. Different reviewer has different commands.

Reviewer 3 Report

Thanks for the revision. The manuscript can be recommended for publication in its present form.

Author Response

Thank you for your reviewing.